# *NTRK* Therapy among Different Types of Cancers, Review and Future Perspectives

**DOI:** 10.3390/ijms25042366

**Published:** 2024-02-17

**Authors:** Nyein Wint Yee Theik, Meri Muminovic, Andres M. Alvarez-Pinzon, Ahmed Shoreibah, Atif M. Hussein, Luis E. Raez

**Affiliations:** 1Division of Internal Medicine, Memorial Healthcare System, Pembroke Pines, FL 33028, USA; ntheik@mhs.net (N.W.Y.T.); ashoreibah@mhs.net (A.S.); 2Memorial Cancer Institute, Memorial Healthcare System, Pembroke Pines, FL 33028, USA; mmuminovic@mhs.net; 3Memorial Cancer Institute, Office of Human Research, Florida Atlantic University (FAU), Pembroke Pines, FL 33028, USA; 4Memorial Cancer Institute, Memorial Healthcare System, Florida Atlantic University (FAU), Pembroke Pines, FL 33028, USA; ahussein@mhs.net

**Keywords:** *NTRK* gene fusion, *NTRK* inhibitor, malignancies

## Abstract

Neurotrophic tyrosine receptor kinase (*NTRK*) has been a remarkable therapeutic target for treating different malignancies, playing an essential role in oncogenic signaling pathways. Groundbreaking trials like NAVIGATE led to the approval of *NTRK* inhibitors by the Food and Drug Administration (FDA) to treat different malignancies, significantly impacting current oncology treatment. Accurate detection of *NTRK* gene fusion becomes very important for possible targeted therapy. Various methods to detect *NTRK* gene fusion have been applied widely based on sensitivity, specificity, and accessibility. The utility of different tests in clinical practice is discussed in this study by providing insights into their effectiveness in targeting patients who may benefit from therapy. Widespread use of *NTRK* inhibitors in different malignancies could remain limited due to resistance mechanisms that cause challenges to medication efficacy in addition to common side effects of the medications. This review provides a succinct overview of the application of *NTRK* inhibitors in various types of cancer by emphasizing the critical clinical significance of *NTRK* fusion gene detection. The discussion also provides a solid foundation for understanding the current challenges and potential changes for improving the efficacy of *NTRK* inhibitor therapy to treat different malignancies.

## 1. Introduction and Epidemiology

Neurotrophic tyrosine receptor kinase (*NTRK*) inhibitors for certain tumors expressing these fusion proteins have recently become a new favored treatment [1]. The *NTRK* genes, including *NTRK* 1, 2, and 3, encode tropomyosin receptor kinase (TRK) receptor family proteins like TRKA, TRKB, and TRKC, known as TRK, responsible for neural cell progress and activities [2]. Any variations or rearrangements of the genes could lead to the activation of neural cells [3]. Among the *NTRK* genes, *NTRK*1 was initially recognized as an actionable oncogene in colon cancer by Mariano Barbacid and colleagues in 1982 [4]. Different *NTRK* translocations or gene fusions and TRK protein overexpression forms were eventually observed in solid tumors besides colon cancer, including other gastrointestinal, gynecological, thyroid, lung, and pediatric malignancies [5].

Among a total of 55 patients in phase 2 of the NAVIGATE trial, 55% of individuals remained progression-free after one year of follow-up, which demonstrated an excellent overall response rate (ORR) to larotrectinib (75% with a 95% confidence interval [CI] of 61–85%) [6]. This groundbreaking trial led to a first-generation *NTRK* inhibitor (larotrectinib) being approved by the Food and Drug Administration (FDA) that is histology agnostic, including treatment for adult and child malignancies. After that, another first-generation inhibitor (entrectinib) became the second inhibitor approved by the FDA [7]. 

According to the phase-2 NAVIGATE trial and phase-1 LOXO-TRK-14001 trial, first-generation *NTRK* inhibitors were found to have a rapid response with effectiveness and survival benefit for the patients with a two-year progression-free survival (PFS) of 67% (95% CI 44–90%) [6]. Therefore, recognizing *NTRK* gene fusions in various solid tumors is crucial for future cancer management, potentially extending the survival of affected patients.

## 2. Different Methods for *NTRK* Gene Fusion Detection

Diverse methods exist for detecting aberrant *NTRK* gene fusions. Each test has its characteristics, limitations, and benefits; however, the choice of test is typically determined by sample diversity and the test’s sensitivity and specificity. One molecular cytogenetics technique called fluorescence in situ hybridization (FISH) employs fluorescently labeled deoxyribonucleic acid (DNA) probes to detect abnormal rearrangements in fusion genes. Formalin-fixed, paraffin-embedded (FFPE) tumor specimens are typically used to visualize aberrant fusion genes. FISH is frequently used to diagnose certain cancers, such as secretory breast carcinoma (SBC) and infantile fibrosarcoma, due to the prevalence of *NTRK* genes in these conditions [8].

Reverse transcriptase polymerase chain reaction (RT-PCR) is generally helpful in detecting *NTRK* fusions by amplifying ribonucleic acid (RNA) molecules and is applied as a confirmation test after other tests show positive results for abnormal expression [9]. One of the most frequently used tests, Next-Generation Sequencing (NGS), is based on known *NTRK* fusion partners, validating the identification of fusion events. The benefit of the test is that it warrants comprehensive genomic profiling and can detect other actionable genetic alterations in addition to *NTRK* fusions and their various variants. Additionally, NGS is commonly used for comprehensive genomic profiling that includes these genes [10].

Immunohistochemistry (IHC) is one of the most widely used and accessible assays due to its low cost and ability to detect abnormal *NTRK* protein expressions. The specificity and sensitivity are usually gene partner-dependent and tumor-dependent, making it an excellent screening test overall. However, confirmation tests are occasionally advised because its value is limited to screening [11]. Total transcriptome RNA sequencing necessitates sequencing the entire RNA transcriptome, which enables the identification of aberrant fusion events based on significant alignment patterns and breakpoints, even though it is pricey and laborious, and RNA technology is less commonly used than DNA NGS.

In summary, DNA NGS in tissue is the best technology to detect *NTRK* fusions, and RNA NGS is the best complement to identify these fusions. Liquid biopsies are an alternative to DNA NGS; however, RNA NGS is not yet widely available in liquid biopsies.

## 3. TRK Biology and Ontogenesis

*NTRK* belongs to the tyrosine receptor kinase family, which includes *NTRK*1, *NTRK*2, and *NTRK*3, encoding for TRKA, TRKB, and TRKC, respectively. *NTRK*1 was initially identified as an oncogene in 1982 by Mariano Barbacid and colleagues during gene transfer assays aimed at discovering genes with transforming capabilities in human tumor specimens, specifically those from colon cancer [4]. Subsequently, the cDNA of the *NTRK*1 proto-oncogene was isolated and characterized as TRKA, a 790-amino acid protein displaying features characteristic of cell surface receptor tyrosine kinases, in 1989 [12]. TRKA was found to be expressed in the nervous system in 1991 and recognized as a receptor for neurotrophic nerve growth factor (NGF) [13]. Additionally, TRKB and TRKC were considered members of the same family of receptors, and all TRK receptors and *NTRK*s play a vital role in nervous system development, differentiation, and apoptosis.

TRK receptors are capable of binding to different ligands such as NGF for TRKA, brain-derived neurotrophic factor (BDNF) or neurotrophin 4 (NT-4) for TRKB, and neurotrophin 3 (NT-3) for TRKC [14]. Neurotrophins were initially identified as survival molecules for neurons. The binding of neurotrophins to their alternative receptor, p75NTR, primarily results in the activation of JNK signaling cascades, as well as the P75NTR-interacting protein (NRAGE, also known as melanoma-associated antigen D1), and P75NTR-associated cell death executor (NADE) adaptors that promote cell cycle arrest and apoptosis directly [15,16,17].

Valent et al. first mapped *NTRK*1, *NTRK*2, and *NTRK*3 to human chromosomes 1q22, 9q22, and 15q25 by fluorescence in situ hybridization in 1997 [18]. In 2012, key regions for achieving selective inhibition were recognized from TRKA and TRKB crystal structures [19]. Although *NTRK* was identified as an oncogene in 1982, the actual development of *NTRK* inhibitors began in 2015. The overall timeline of *NTRK* genes and inhibitor therapy is summarized in Table 1.

TRK receptors can be activated by different mechanisms, such as the G-protein coupled receptor (GPCR) mechanism, without neurotrophins’ involvement. In order to increase the survival of neural cells through AKT activity, the TRK receptor can be activated by two GPCR ligands, adenosine, and pituitary adenylate cyclase-activating polypeptide (PACAP) [20]. Somatic *NTRK* mutation has been identified in various tumor types, including colorectal and lung cancers such as large cell neuroendocrine carcinoma and NSCLC, as well as melanoma and acute myeloid leukemia [21,22,23,24].

## 4. Mechanisms of Action 

Both first-generation *NTRK* inhibitors are orally administered and warranted to treat adult solid malignancies and some pediatric tumors. They bind selectively to the ATP-binding site of abnormal *NTRK* gene fusion proteins, inhibit ATP from binding to the fusion protein, and inhibit the action of signals in its downward path that collaborate in the growth of cells, survival, and differentiation. This inhibition disrupts the disruption of abnormal signaling cascades and encourages tumor regression (Figure 1) [25].

Correspondingly, both first-generation inhibitors, entrectinib and larotrectinib, are highly effective in inhibiting the growth of BaF3 cells transduced with different *NTRK* fusions and of primary cancer cell lines harboring *NTRK* rearrangements in vitro and in mice via inhibition of MAPK, PI3K-AKT, PKC, and STAT3 pathways [26]. *NTRK* inhibitors have shown various advantages for patients with tyrosine kinase tumors, with practical and long-lasting responses approved for adult and pediatric populations [27].

## 5. First-Generation *NTRK* Inhibitors

### 5.1. Larotrectinib

First-generation *NTRK* inhibitor trials predominantly adopt a basket trial design due to the infrequent occurrence of *NTRK* fusions within specific tumor types. Larotrectinib, the inaugural pan-TRK inhibitor, selectively targets TRK receptor proteins, namely TRKA, TRKB, and TRKC. Three pivotal clinical trials, namely LOXO-TRK-14001 (NCT02122913), SCOUT (NCT02637687), and NAVIGATE (NCT02576431), collectively enrolled 55 eligible patients across 21 distinct cancer types [6,28,29]. The cohort comprised both adult and pediatric patients administered with 100 mg of larotrectinib twice daily. The objective response rate (ORR) was calculated at 75% (with a 95% confidence interval of 67–85%) [6]. Notable adverse effects, such as gastrointestinal and central nervous system manifestations like dizziness, were observed; however, the majority were graded as mild (Grade 1) and manageable. An updated analysis conducted in 2021, encompassing a total of 218 samples, reported an ORR of 75%, comprising a complete response (CR) rate of 22%, a partial response (PR) rate of 53%, stable disease (SD) in 16% of cases, and progressive disease (PD) in 6%. Larotrectinib received approval for the treatment of *NTRK* gene fusion tumors in both adult and pediatric populations in November 2018. Notably, most treatment-related adverse events were categorized as Grade 1–2, with only 18% exhibiting Grade 3–4 severity in subsequent investigations [30]. Grade 3 events included myalgia, hypersensitivity reactions, and weight gain, with only 2% of patients requiring discontinuation of larotrectinib due to side effects, thus underscoring its favorable safety profile as a therapeutic option.

### 5.2. Entrectinib

Entrectinib is a multikinase inhibitor that halts reactive oxygen species (ROS) oncogene 1 (ROS1) and anaplastic lymphoma kinase (ALK) in addition to blocking tyrosine receptor kinase A, B, and C [31]. Integrated analysis of four trials in 2019 included STARTRK-1, STARTRK-2, and ALKA-372-001, with 54 adult patients aged over 18 years who received entrectinib [32]. Common toxicities are noted in Grades 1 and 2, including neurologic complications such as dizziness (27% of patients), cognitive changes (4%), and weight gain, which is most commonly reported as Grade 3. No treatment-related mortality was noted. The study revealed a CR of 7% and a PR of 50%. Based on the report, the FDA approved entrectinib for adult and pediatric populations in August 2019, and subsequently, the European Medicines Agency also approved entrectinib in 2020 due to its safety profile [33]. 

## 6. Second-Generation *NTRK* Inhibitors

### 6.1. Repotrectinib

Due to the limited durability and resistant mutations against first-generation inhibitors, second-generation inhibitor trials have started to overcome the difficulties. Repotrectinib has a smaller molecular weight and a more compact macrocyclic structure than the first generation [34]. Repotrectinib is a small compound compared to other inhibitors created to accommodate the bulky, positively charged arginine side chain in the solvent front without steric clashes. In phases 1 and 2 of the TRIDENT-1 study (NCT03093116), with 40 populations who received repotrectinib, the ORR was 41–62% [35]. Common side effects from the study include mild Grade 1 ataxia, paresthesia, nausea, perioral numbness, and dysgeusia—the FDA approved repotrectinib to treat patients with advanced solid tumors due to the safety profile of the medication.

### 6.2. Selitrectinib

Although the major clinical trials of selitretinib, including NCT03206931 and NT03215511 are ongoing, most case reports reported that selitrectinib might be an efficient medication for solid malignancies [36,37]. Additionally, patients resistant to entrectinib through the *NTRK*3 G623R mutation revealed good responses with selitrectinib per the 2021 report [38]. 

### 6.3. Taletrectinib

Taletrectinib is a newly emerged ROS1/*NTRK* kinase inhibitor with high activity against the G2031R solvent-front mutation compared to other inhibitors [39]. According to the published report of the first human phase 1 study (TRUST) in 2020 with 46 patient populations, the confirmed objective response rate was 33.3% [40]. Most common treatment-related adverse events, such as nausea (47.8%), diarrhea (43.5%), and vomiting (32.6%), were noted. However, there is no clinical data on the effect of taletrectinib on overcoming resistance to first-generation inhibitors. Table 2 summarizes the overall clinical trials, response results, and commonly noted side effects per trial.

### 6.4. Other Agents

Several multi-target tyrosine kinase inhibitors (TKIs) with varying inhibitory activity against tropomyosin receptor kinase (TRK) have received approvals for indications beyond the treatment of patients with *NTRK* fusions. Crizotinib, initially designed as a MET inhibitor, was later recognized as an inhibitor of anaplastic lymphoma kinase (ALK), ROS1, and TRK. However, its affinity for TRK is significantly lower than for MET, ALK, and ROS1. It is approved for treating ALK- and ROS1-rearranged non-small cell lung cancer (NSCLC) [41].

Cabozantinib, approved by the FDA for renal cell carcinoma and medullary thyroid carcinoma, targets multiple receptor tyrosine kinases, including MET, RET, AXL, TRKA, and TRKB [42,43,44]. Ponatinib, originally developed as a BCR-ABL1 inhibitor for chronic myelogenous leukemia, has demonstrated efficacy against most BCR-ABL1 resistance mutations and has shown potential in suppressing the growth of *NTRK* fusion-positive tumors in preclinical trials [45,46].

Nintedanib, recognized as an anti-angiogenic drug and vascular endothelial growth factor receptor (VEGFR) tyrosine kinase inhibitor, has also exhibited inhibitory effects on platelet-derived growth factor receptor (PDGFR), fibroblast growth factor receptor (FGFR), and TRK kinases [47,48].

The clinical activity of these multi-target TRK inhibitors in patients with *NTRK* fusion-positive cancers is not extensively characterized. However, well-established developments in larotrectinib, entrectinib, and second-generation inhibitors like repotrectinib and taletrectinib provide promising avenues for treating cancers harboring *NTRK* fusions. 

### 6.5. Mechanisms of Resistance to NTRK Inhibitors

However, resistance to first-generation inhibitors was identified through the mutations of *NTRK*1 gene fusion proteins such as G667C and G595R [49]. In a 2021 report, one patient with mammary secretory carcinoma of the parotic gland and the ETV6-*NTRK*3 fusion reported secondary resistance to entrectinib through the *NTRK*3 G623R mutation [38]. Second-generation inhibitors, which include selitrectinib and repotrectinib, have emerged to overcome the mechanism of acquired resistance. However, recent trials indicated that resistance against the second generation develops through the mutation of TRK proteins such as xDFG [50]. 

### 6.6. Overall Side Effects and Effects of TRK Inhibition

According to Smeyne RJ et al., in a 1994 trial, *NTRK*1-null mice lacking most sympathetic neurons did not exhibit nociceptive and temperature sensations and eventually passed away within a month due to severe sensory and sympathetic neuropathies [51]. In comparison, *NTRK*2 knockout mice, lacking motor neurons or dorsal root and trigeminal ganglia neurons, died perinatally due to a lack of eating [52]. *NTRK*3-null mice also exhibited deficits in the quality and quantity of motor neurons and deficits in a population of dorsal root ganglia neurons, resulting in abnormal movements and posture [53]. Homozygous disruption of the *NTRK*2 gene revealed an increase in apoptosis of endothelial cells and a decreased number of intramyocardial blood vessels. In contrast, targeted deletion of all TRKC isoforms in mice led to severe cardiac deficiencies, including atrial, ventricular, and valvular defects, resulting in animal death in the early postnatal period [54].

However, according to clinical studies conducted in humans, it is remarkable that first- and second-generation *NTRK* inhibitors are very well-tolerated. Side effects are usually manageable and *NTRK* inhibitors have proven effective in malignancies that are positive for *NTRK* gene fusion. The most common side effects were typically gastrointestinal, and 30% of patients developed dizziness, in addition to other central nervous system side effects like ataxia, paresthesia, and perioral numbness. According to the study by Qin et al., 10% of 218 patients displayed weight gain, and 12% developed anemia [55]. Other non-specific side effects, such as fatigue and elevated liver enzymes, are also commonly noted among patients treated with *NTRK* inhibitors. Importantly, the mentioned side effects are known to be reversible after discontinuing the medications.

## 7. Role of *NTRK* Inhibitors in Various Types of Cancers and Adverse Effects

*NTRK* gene fusions have been identified across a spectrum of malignancies, encompassing diverse cancer types such as lung cancer, melanoma, colorectal cancer, thyroid cancer, and leukemia. In contrast to rare cancers, where *NTRK* gene alterations are detected in approximately 80% of cases, the prevalence of gene fusions in common solid malignancies is markedly lower, ranging from 5 to 25%. Notably, in prevalent solid tumors like lung and breast cancers, the frequency of *NTRK* gene fusions is less than 1% [56].

Rare tumors exhibiting a substantial prevalence of *NTRK* fusion genes include carcinoma ex-pleomorphic adenoma, secretory breast carcinoma, congenital mesoblastic nephroma, and infantile fibrosarcoma. Despite the infrequency of *NTRK* gene fusions in common solid tumors, their identification holds paramount importance due to the emerging role of *NTRK* inhibitors in targeted therapeutic interventions.

This review study aims to comprehensively discuss recent applications of *NTRK* inhibitor therapy in selected common malignancies. By exploring the therapeutic landscape of *NTRK* inhibitors in prevalent cancers, we endeavor to contribute to the evolving understanding of their efficacy and potential implications in oncology practice.

### 7.1. NTRK Gene Fusion in Lung Cancer

Among lung cancers, especially non-small cell lung cancer (NSCLC), *NTRK* gene fusions can be found only 0.5% of the time. However, gene fusion can also be found in other subtypes like neuroendocrine carcinoma and sarcomatoid tumors of the lung [57]. *NTRK*1 and *NTRK*2 are commonly expressed in squamous cell carcinoma compared to adenocarcinoma and small cell lung cancer. First-generation inhibitors have been proven to be a practical and durable therapy for managing NSCLC. 

For example, according to the Drilon et al. trial, which published results in 20 patients with *NTRK* fusion-positive lung cancer, the ORR among 15 evaluable patients evaluated by investigator assessment was 73% (95% CI, 45–92); the CR was 7%, the PR was 67%, 20% had stable disease, and 7% had progressive disease as the best response [58]. The median duration of response, PFS, and overall survival were a total of 33.9 months (95% CI, 5.6–33.9), 35.4 months (95% CI, 5.3–35.4), and 40.7 months (95% CI, 17.2-not estimable), respectively. The ORR among patients with CNS metastases at baseline was 63% (95% CI: 25–91) [58]. In a recent report about ctDNA analysis in 2023 with a patient treated with larotrectinib in two clinical trials, the ORR was 83% (95% CI 61–95) with an extended survival benefit and a favorable safety profile in a patient with advanced lung cancer [59].

Most adverse events were considered as Grade 1 or 2. Nevertheless, resistance may develop due to fusion variants such as TPM3-*NTRK*1. Similarly, EGFR-TKI and gene fusion partners, including ETV6 and SQTTM1, have demonstrated resistance to third-generation inhibitors [60].

### 7.2. NTRK Gene Fusion in Colorectal Cancer

In approximately 0.7% of colon cancers, *NTRK* gene fusion, especially *NTRK*1 and *NTRK*3, is found alongside APC and TP53 gene aberrations and RAS/BRAF gene alterations [61]. The TPM3-*NTRK*1 rearrangement is commonly seen [62]. *NTRK* is prevalent in females with a right-sided primary colorectal tumor, RAS/RAF WT status, and MSI phenotype, according to data from the phase 2 NAVIGATE trial, which produces various responses in patients to larotrectinib with the gene-positive fusion, and which is locally advanced, and in metastatic GI cancers, especially in colorectal cancer with an ORR which is 3% and a partial response rate of 30% [63,64]. Therefore, it is crucial to identify *NTRK* fusions in GI malignancy patients, especially those with high microsatellite instability rates [65].

### 7.3. NTRK Gene Fusion in Central Nervous System Malignancies

Central Nervous System (CNS) tumors, such as low- and high-grade gliomas, are distinguished from other solid malignancies by the quantity of *NTRK* gene fusion [66]. The *NTRK*1/TRKA fusion has been linked to an improved prognosis and an increased likelihood of tumor regression. The *NTRK*2/TRKB fusion has been found to be detrimental in certain tumors [66]. Gliomas, characterized by *NTRK* gene expression, tend to affect multiple regions of the nervous system, particularly the hemisphere and frequently have an aggressive clinical course. The cohort study conducted by Torre et al. identified the detailed anatomical distribution of *NTRK* fusions, revealing a predominant occurrence in the hemispheric region (66.7%). Additionally, *NTRK* fusions were observed in the brainstem (9.5%), cerebellum (7.1%), and optic nerve/suprasellar region/deep grey nuclei (4.8%). However, among pediatric *NTRK*-fused gliomas, the distribution appears to be more diverse [67].

In CNS tumors, the frequency of *NTRK*2 (11% of GBM) and *NTRK*1 (1% of GBM) fusions exceeds that of *NTRK*3. Notably, first-generation inhibitors have exhibited the ability to cross the blood–brain barrier, with larotrectinib demonstrating remarkable specificity for *NTRK* fusions and notable efficacy against CNS tumors [68]. Given the limitations of surgical resection and radiotherapy in treating brain malignancies, identifying oncogenes such as *NTRK* gene fusions becomes crucial. The identification and targeting of *NTRK* fusions may offer critical clinical advantages [69]. Additional research in cancer neuroscience and clinical trials is required to explore the full potential of *NTRK*-targeted therapies in effectively treating CNS cancers.

### 7.4. NTRK Gene Fusion in Thyroid Cancers

Thyroid cancer, like other solid tumors, has few *NTRK* fusions. *NTRK* rearrangement in the TKD of the *NTRK*1 gene can arise in thyroid cancer, particularly papillary thyroid carcinoma. Furthermore, a gene mutation known as the ETV-*NTRK*3 fusion is present in around 14.5% of patients diagnosed with thyroid cancer due to radiation exposure following the Chornobyl nuclear accident [70]. IRF2BP2-*NTRK*3 is also spotted in papillary thyroid carcinoma [61]. In a report in 2023 combining the three clinical trials with larotrectinib (NCT02576431, NCT02122913, NCT02637687) with 30 adult patients with fusion-positive thyroid cancers, the ORR was 63% (95% CI 44–80%), the CR was 53%, the PR was 17%, and 13% had progressive disease [6,28,29,71]. Larotrectinib produced a rapid and durable response, extended survival, and a favorable safety profile in this trial. 

### 7.5. NTRK Gene Fusion in Hematological Malignancies

Hematological malignancies, including acute myeloid and lymphoblastic leukemia, chronic lymphocytic leukemia (CLL), and Philadelphia-positive ALL, frequently have the *NTRK*-ETV6 fusion. Taylor et al. reported that larotrectinib, a first-generation inhibitor, reduced expression of the fusion gene while inhibiting TRK activity in patients with *NTRK*2-ETV6 variants associated with histiocytosis and complex myeloma [72]. 

Notably, TP53 cell lines are more sensitive to the first-generation inhibitor [73]. Despite these promising results, TRK aberrations have been shown to have reactivity against the therapeutic target, limiting the efficacy of clinical benefits in certain hematological tumors. Further studies and research are needed to fully understand the potential clinical benefits of *NTRK* therapy in various hematological malignancies.

### 7.6. NTRK Gene Fusion in Sarcomas

Among soft tissue sarcomas, the predominant characterization of infantile fibrosarcoma and lipofibromatosis-like neural tumors revolves around ETV6-*NTRK*3 fusions and *NTRK*1 gene rearrangements [74,75]. In a minority of cases, *NTRK*3-negative infantile fibrosarcomas have been documented to exhibit *NTRK*1 gene rearrangements instead [76]. Chiang et al.’s research highlights the occurrence of *NTRK* gene rearrangements within a subset of undifferentiated uterine sarcomas displaying fibrosarcoma-like morphology [77]. Kojima et al. also used immunohistochemistry to discover CD30 expression in mesenchymal tumors with kinase gene fusions, including *NTRK*-rearranged tumors and BRAF, RAF1, or RET fusion tumors [78]. Xu et al. highlighted molecular testing for kinase fusions in spindle cell neoplasms exhibiting specific features, emphasizing the significance of pan-TRK positivity in *NTRK* fusions [79]. Incorporating the *NTRK* fusion into the existing diagnostic evaluation of sarcoma patients presents a unique challenge due to the biomarker’s rarity. IHC serves as a valuable initial screening tool, and implementing targeted MPS panels, specifically designed to identify *NTRK* gene fusions, in routine genome-wide MPS for sarcomas is cost-effective, given the limited number of additional genomic alterations requiring testing [80].

### 7.7. NTRK Gene Fusion in Melanocytic Tumors

*NTRK* fusions have a diagnostic and classification impact among melanocytic tumors. Among melanocytic neoplasms, Spitz tumors are one subgroup with distinct morphological features that tend to affect young individuals [81,82]. Yeh et al. identified ETV6-*NTRK*3, MYO5A-*NTRK*3, and MYH9-*NTRK*3 fusions in Spitz tumors [83]. Additionally, Yeh et al. report 38 Spitz tumors with *NTRK*1 fusions with distinctive histopathologic features which is helpful in diagnosis and helps prioritize case selection for molecular testing in patients that need targeted therapy [84]. Atypical Spitz tumors (AST) deviate from stereotypical Spitz tumors due to atypical features and are now regarded as an intermediate category of melanocytic tumors with uncertain malignant potential. Cappellesso et al., with a series of 180 AST screens with pan-TRK IHC and confirmed with FISH, detected *NTRK* 1 and *NTRK* 3 fusions [85]. 

Among melanocytic tumors, acral and mucosal melanomas are aggressive subtypes with a significantly lower somatic mutation burden than cutaneous melanomas but more frequent genetic variations, focused gene amplifications, and structural alterations. *NTRK* fusions are common in spitzoid melanoma, with a prevalence of 21–29% compared to <1% in cutaneous or mucosal melanoma and 2.5% in acral melanoma, which reveals that fusion proteins are mutually exclusive for most common oncogenic drivers such as BRAF or NRAS [86]. The frequency of the *NTRK* fusion gene, *NTRK*1, *NTRK*2, and *NTRK*3 in metastatic non-acral cutaneous melanoma is approximately 0.8% and 0.9% in mucosal and para-mucosal melanomas [87]. The meta-analysis by Wang et al. focuses on different gene mutations such as MITF, PTEN, ATM, PRKN, and BRAF V600E mutations in sacral melanoma by shedding new light on the pathogenesis and broadening the catalog of therapeutic targets for this difficult-to-treat cancer [88].

### 7.8. NTRK Gene Fusion in Salivary Gland Tumors

The presence of *NTRK* gene fusions has been specifically observed in rare tumors, particularly in secretory carcinomas of the breast and salivary gland, indicating a unique prevalence. The most common fusion variant within the category of salivary gland carcinomas is ETV6-*NTRK*3. This fusion subtype has been consistently identified in different types of salivary gland carcinomas, including two cases of secretory carcinoma and one case of adenocarcinoma [89]. The frequent presence of the ETV6-*NTRK*3 fusion in salivary gland tumors highlights its importance as a possible diagnostic indicator and target for medical treatment. In order to gain a better understanding of the complex molecular mechanisms and medical significance of *NTRK* gene fusions in salivary gland tumors, it is essential to conduct extensive clinical research and molecular profiling studies.

### 7.9. Other Malignancies

The fusion of *NTRK*3 and ETV6 is mainly linked to around 90% of instances of rare secretory breast carcinoma, along with a specific group of triple-negative breast cancer subtypes [90]. Moreover, *NTRK* fusion genes exhibit a notable prevalence in specific pediatric malignancies such as congenital melanocytic nevus (CMN) and secretory breast carcinoma (SBC). Irrespective of tumor histology, the therapeutic response to first-generation inhibitors has demonstrated remarkable efficacy, with a superior response rate of 75% observed in 73% of cases within a 6-month timeframe, particularly with entrectinib. Notably, although slightly reduced compared to the 6-month response, an overall response rate (ORR) of 57% has been documented in 68% of patients [6]. These findings underscore the broad therapeutic potential of targeting *NTRK* fusion genes across diverse malignancies and highlight the promising clinical outcomes achievable with targeted *NTRK* inhibition strategies. Additional research is necessary to better understand the mechanisms underlying the response to *NTRK* inhibitors and the optimization of treatment protocols are warranted to maximize therapeutic efficacy and patient outcomes.

### 7.10. Future Directions and Challenges

Beginning with the NAVIGATE trials, multiple clinical trials have demonstrated the greater efficacy and superior outcomes of *NTRK* inhibitors in different types of solid tumors and hematological malignancies. Due to the rarity of *NTRK* gene fusions in most malignancies, testing for it may be challenging. In addition, selecting reliable testing may be difficult due to sensitivity and specificity issues with specific tests, which necessitate confirmation tests, sample limitations, and the expense of the tests. 

However, patients with *NTRK* mutations and amplifications cannot receive inhibitor therapy due to other non-actionable genetic aberrations [91]. *NTRK* inhibitor therapy could be costly and difficult to evaluate in some regions. On the other hand, we expect the cost to be reasonable based on the required dose compared to conventional chemotherapy for specific cancer types. Moreover, a large generalized clinical trial might be required to investigate details of long-term side effects, estimate the total duration of therapy, and see whether it is beneficial with combined treatment or single treatment. 

Developing more predictive and sensitive biomarkers could also be helpful in further tests for gene fusion. Despite that, efforts have been made to expand patients’ access to genetic testing and targeted therapies to ensure equitable and widespread implementation. Regardless, *NTRK* therapy has shown successful outcomes in many clinical trials, highlighting the critical aspect of comprehensive genomic profiling in cancer patients. In the future, the treatment will be applied widely.

## 8. Conclusions

*NTRK* therapy has emerged as a promising treatment option for cancer patients with *NTRK* fusion-positive tumors. Following the landmark NAVIGATE clinical trials, the development of first- and second-generation *NTRK* inhibitors transformed the treatment landscape, providing superior clinical outcomes, greater quality of life, and longer survival rates than traditional TKIs. The recommendation for outpatient settings is to use comprehensive nucleic acid-based profiling and complementary immunohistochemistry (IHC) assays to detect abnormal gene fusions. However, issues such as limited access to testing and associated treatment costs continue. Despite these challenges, there is a compelling need for additional clinical trials to investigate the long-term side effects and comparative efficacy of *NTRK* therapy versus conventional chemotherapy or combination therapies. These endeavors are critical for comprehending the overall potential of *NTRK* therapy while addressing existing healthcare system barriers. Such initiatives represent a paradigm shift in precision medicine and offer optimism towards the future of cancer treatment.

## Figures and Tables

**Figure 1 ijms-25-02366-f001:**
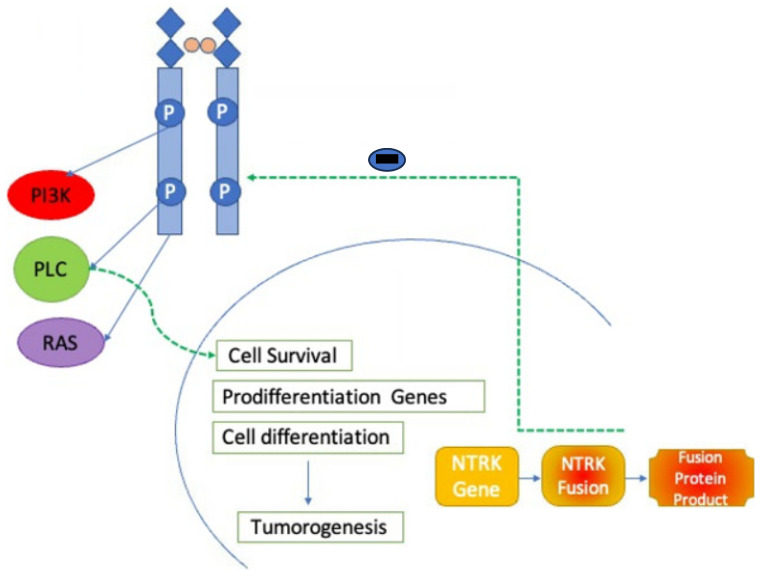
*NTRK* inhibitor mechanism of action. *NTRK* inhibitors block action through the *NTRK* receptor preventing downstream activation of phoshoinositide-3-kinase (PI3K), phospholipase C (PLC), and RAS pathways.

**Table 1 ijms-25-02366-t001:** Timeline in year of *NTRK* gene and inhibitor therapy development.

1982	Identification of *NTRK* as an oncogene in patient with colorectal carcinoma
1989	Isolation of cDNA of the *NTRK*1 proto-oncogene
1997	Gene mapping of *NTRK*1, *NTRK*2, and *NTRK*3 to human chromosomes, 1q22, 9q22, and 15q25 by FISH
2015	First-generation *NTRK* inhibitors entered clinical trials
2017	Second-generation *NTRK* inhibitors entered clinical trials
2018	FDA granted accelerated approval for larotrectinib for adult and pediatric patients with *NTRK* fusion-positive solid tumors
2019	FDA granted accelerated approval for entrectinib for adult and pediatric patients with solid tumors with *NTRK* gene fusion without a known acquired resistance mutation
2020	FDA granted Fast Track to repotrectinib in *NTRK*-positive advanced solid tumors
2021	Phase 2 basket trial of taletrectinib for solid tumors with *NTRK* initiated

**Table 2 ijms-25-02366-t002:** Summarization of the clinical trials, findings, and side effects of *NTRK* inhibitors.

Medication	Target Genes	Related Trials	Findings	Side Effects
Larotrectinib	TRKA/B/C	NCT02576431 NCT02122913 NCT02637687	ORR 75% (95% CI 68–81) CR 22% PR 109% Stable 16% PD 6%	18% Grade 3–4 treatment-related side effects
Entrectinib	TRKA/B/C, ROS1, ALK	ALKA-372-001 STARTRK-1 STARTRK-2	ORR 57% CR 7% PR 50% Stable 17% PD 7%	10% weight gain, 12% anemia, 4% CNS manifestation
Repotrectinib	TRKA/B/C, ROS1, ALK	NCT03093116 (TRIDENT-1)	ORR 41–62%	Grade 1 CNS-related side effects
Selitrectinib	TRKA/B/C	NCT03215511 NCT03206931	N/A	N/A
Taletrectinib	TRKA/B/C, ALK	NCT04395677	N/A	GI-related side effects—nausea, diarrhea, vomiting

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
