# Peer review of "NTRK Therapy among Different Types of Cancers, Review and Future Perspectives"

_ijms, 2024, doi:10.3390/ijms25042366_

Round 1
Reviewer 1 Report
Comments and Suggestions for Authors
The authors provided a comprehensive review on Neurotrophic tyrosine receptor kinase (NTRK) inhibitors. My only suggestion is to include a figure with details about the NTRK signalling pathway and include inhibitor’s targets described in sections 5 and 6. This would make easier for the reader to visualize which part of the pathway are being affected by the inhibitors.
Author Response
Thank you very much for reviewing and provided feedback to improve. For section 5 and 6, we added table for each inhibitor's targets. After discussing with my fellow colleagues and mentor's suggestions, we believe adding more figures won't be benefit to paper. Thank you again and have a wonderful day!
Reviewer 2 Report
Comments and Suggestions for Authors
In the manuscript by Theik et al., the authors review the importance of NTRK fusion-positive malignancies and the therapeutic benefit and potential of NTRK inhibitors. Overall, the manuscript is succinct while still informative, and generally well written. Minor comments below.
- In section 7.3, the first sentence is somewhat redundant as written. It would ready better, and more correctly, as follows: "Central Nervous System (CNS) tumors, such as low- and high-grade gliomas, are distinguished from other solid malignancies by the quantity of NTRK gene fusion (62)."
- Lines 291-292. It is not correct that "KCTD16-NTRK1 is usually present in ganglioglioma". NTRK fusions are occasionally encountered in ganglioglioma, but this particular tumor is more likely to harbor BRAF V600E.
- In Section 7.3, the authors should cite and discuss Torre et al. 2020 (PMID 32665022), as this is an important clinicopathologic study examining the presence of NTRK fusions in gliomas.
- Line 99. The word "Key" is incorrectly capitalized.
Comments on the Quality of English LanguageOverall fine, needs minor proof checking.
Author Response
Thank you very much for providing detailed feedback!
-For section 7.3 feedback, we changed the sentence per suggestion.
-Line 291-292 feedback, we removed the original sentence and corrected it.
-Section 7.3, added references per suggestion
-Word (Key) corrected
Thank you again for the time to review and have a wonderful day
Reviewer 3 Report
Comments and Suggestions for Authors
Nyein Wint Yee Theik et al present an interesting and well-written review on the role of NTRK-inhibitors, for which I suggest the acceptance after the minor revision:
1-Please, expand and better detail the chapter on NTRK-fused sarcomas, a category that almost daily sees new and drastic “classificatory news”. Please, refer to the following literature (doi: 10.1097/PAS.0000000000001055; doi: 10.1016/j.modpat.2022.100083; doi: 10.1016/j.annonc.2020.08.2232; doi: 10.1097/PAS.0000000000001982). In the present form, this chapter is too confounding and few informative;
2-Please, add a separate chapter for melanocytic tumors. In this tumors, NTRK fusions (probably less relevant from a therapeutic point of view) have an incredible diagnostic and classification impact, as they define (almost always) a specific subgroup of melanocytic lesions and specifically of Spitz tumors (doi: 10.3390/ijms222212332; doi: 10.1097/PAS.0000000000001294; doi: 10.1002/path.4775; doi: 10.1097/PAS.0000000000001235; doi: 10.1177/1066896916630375). Please, specify the prevalence and pathological meanings of NTRK fusions in acral and mucosal melanoma (doi: 10.1111/ddg.14160; doi: 10.1186/s13073-022-01068-0), before focusing on the so-small subgroup of conventional but “spitzoid” (but not Spitz!!!!) melanocytic tumors with NTRK fusions (your Reference number 73).
3-Please, modify, in the Introduction, the following sentence "However, confirmation tests are typically advised due to their unreliable specificity, and their value is limited to screening". The specificity and sensitivity of this test is “gene partner-dependent” and “tumor-dependent”. However, is often “good overall”, making this test an excellent screening test (doi: 10.1097/PAS.0000000000001294; doi: 10.3390/ijms222212332; doi: 10.3390/ijms23115911; doi: 10.1097/PAS.0000000000000911; doi: 10.1016/j.cancergen.2021.12.010; doi: 10.3390/ijms23137470; doi: 10.1016/j.modpat.2023.100192; doi: 10.1016/j.modpat.2023.100346). Especially in some melanocytic neoplasms, multiple molecular tests are often required to confirm a NTRK fusion identified at immunohistochemistry (doi: 10.3390/ijms222212332; doi: 10.3390/ijms23115911).
4-In several tumors, major attention is being focused on the duplication/gene amplifications (rather than fusion) of kinases and NTRK (doi: https://doi.org/10.23838/pfm.2017.00142). Please, discuss this relvant topic and its potential repercussion in this scenario.
Author Response
Thank you very much for the detailed feedback and comments in addition to sharing some references to add.
-We expanded and added details about Sarcoma section. We appreciated and also added citation that reviewer shared.
-We added and broadly discussed about melanocytic tumor section by added citation that reviewer shared.
-We changed sentence correction according to suggestion.
-We corrected amplication/duplication per original paper
Thank you very much again for the time to review and detailed feedback to our paper. Have a wonderful day!